# A Method for Estimating 24 h Urinary Sodium and Potassium Excretion by Spot Urine Specimen in Stroke Patients

**DOI:** 10.3390/nu14194105

**Published:** 2022-10-02

**Authors:** Beike Wu, Hongmei Yang, Xinyu Ren, Zijing Qi, Shuai Tang, Xuejun Yin, Liping Huang, Maoyi Tian, Yangfeng Wu, Xiangxian Feng, Zhifang Li

**Affiliations:** 1School of Public Health, Shanxi Medical University, Taiyuan 030001, China; 2School of Public Health, Changzhi Medical College, Changzhi 046000, China; 3The George Institute for Global Health, University of New South Wales, Newtown, NSW 2042, Australia; 4School of Public Health, Harbin Medical University, Harbin 150081, China; 5Clinical Research Center, Peking University, Beijing 100088, China

**Keywords:** sodium, potassium, spot urine, 24 h urine, stroke

## Abstract

Spot urine specimens have been used to estimate 24 h urinary sodium (Na) excretion (24UNaV) and potassium (K) excretion (24UKV). However, the validity is limited for 24UNaV and unknown for 24UKV in stroke patients, using the existing formulas. Herein, we developed and validated a new formula for 24UNaV and 24UKV by spot urine specimens in stroke patients. Spot and 24 h urine samples were collected from 970 stroke patients. The models of 24UNaV and 24UKV were developed using stepwise multivariate linear regression in 689 patients. The performance of different formulas was internally validated in 281 patients at the population and individual levels. The obtained new formulas were: (1) estimated 24UNaV (mmol/day): −0.191 × Age + 4.349 × BMI + 0.229 × Spot_Na_ + 1.744 × Spot_Na_/Spot creatinine (Cr) + 41.492 (for male); −1.030 × Age + 2.011 × BMI + 0.143 × Spot_Na_ + 1.035 × Spot_Na_/Spot_Cr_ + 147.159 (for female); and (2) estimated 24UKV (mmol/day): −0.052 × Age + 0.410 × BMI + 0.031 × Spot_K_ + 33.280 × Ln (spot_K_/spot _Cr_) − 5.789 × Ln (spot_Na_/spot _Cr_) − 1.035 (for male); −0.235 × Age + 0.530 × BMI + 0.040 × Spot_K_ + 30.990 × Ln (spot _K/_spot _Cr_) − 7.837 × Ln (spot_Na_/spot_Cr_) + 4.318 (for female). The new formula obtained the lowest mean bias (5.17 mmol/day for 24UNaV and 0.85 mmol/day for 24UKV) and highest proportion at the cutoff under the ±30% level for the estimation of 24UNaV (59.43%) and 24UKV (70.11%). The new formula provides a meaningful exploration to estimate 24UNaV and 24UKV in stroke patients by using spot urine specimens.

## 1. Introduction

Stroke is the second leading cause of death and the third leading cause of disability worldwide [1], and is a major health threat in China [2,3]. Many studies have shown that the risk of stroke is associated with high sodium intake and low potassium intake [4,5,6,7,8,9,10,11,12]. Estimating sodium and potassium intake among stroke patients is an important strategy for improving the intake of both electrolytes and reducing the risk of stroke. The collection of 24 h urine is considered the most reliable method to estimate sodium and potassium intake, both at the individual and population levels [13,14,15]. However, this method has some limitations, such as high costs, high burden for collection, and a high risk of collection errors [16,17]. To overcome these limitations, multiple estimating equations using spot urine specimens have been developed to estimate 24 h urinary sodium excretion (24UNaV) and potassium excretion (24UKV) [18,19]. The commonly validated equations include the Kawasaki formula [13], INTERSALT formula [20], and Tanaka formula [14].

Various prior validation studies indicated that the underlying study population where the validation was conducted is an important factor that affects the performance of these equations [18,21]. For instance, in Asian populations, where salt intake is very high, the Kawasaki equation may perform better than the other equations [22,23,24], and in Western populations, the INTERSALT equations may outperform the others [25]. Even within the INTERSALT study, the intercepts and coefficient parameters of different population characteristics that were included in the equations varied for different regions [20]. Therefore, these previously published equations may not be adequate to estimate the 24UNaV among stroke patients. In addition, there have been much fewer studies for estimating 24UKV compared to estimating 24UNaV.

This study aimed to develop and validate a new simple formula for estimating 24UNaV and 24UKV in stroke patients. In this paper, the performance of the new formula is compared to that of previously published equations.

## 2. Materials and Methods

### 2.1. Study Participants

The Salt Substitute and Stroke Study (SSaSS), a large-scale cluster randomized controlled trial in stroke patients, evaluated the effect of low-sodium/high-potassium salt substitute on the risk of stroke, acute coronary syndrome, and total mortality. This trial was conducted in rural areas of five northern provinces of China over five years (registration number: NCT02092090) [26]. The SSaSS was sponsored by the George Institute for Global Health, Australia. During the SSaSS, spot urine and 24 h urine specimens were collected. A total of 4211 subjects at high risk for cardiovascular and cerebrovascular diseases and 4083 stroke patients from Shanxi province, China were included in the SSaSS. This study was conducted using the data from the SSaSS in Shanxi province.

The study participants were from Gaoping and Changzhi cities of Shanxi province, China. In detail, 6 villages from each city and 20 subjects from each village were randomly selected for subject enrollment. The villages were randomly assigned in a 1:1 ratio to the intervention group, in which the participants used a salt substitute (75% sodium chloride and 25% potassium chloride by mass), or to the control group, in which the participants continued to use regular salt (100% sodium chloride). The intervention was from March 2015 to October 2019, and one spot urine sample and one 24 h urine sample were collected from each subject every 12 months. Inclusion criteria: (1) participants with a history of prior stroke; (2) participants aged 60 years old or above with uncontrolled high blood pressure (BP) (systolic BP ≥ 140 mmHg at visit if on BP lowering medication, or systolic BP ≥ 160 mmHg if not on BP lowering medication). Participants with the following conditions were excluded from the urine collection: (1) urinary incontinence; (2) unable to collect urine on their own and could not find help from others; (3) acute or chronic urinary tract infection, vaginal infection, or perianal infection; (4) acute or chronic urinary tract, vaginal, or gastrointestinal bleeding; (5) females who were pregnant, lactating, or menstruating; (6) vomiting or diarrhea.

The study received ethical approval from the Center for Health Sciences at Peking University in China and the Institutional Review Board at the University of Sydney, Australia. Each study participant signed a written informed consent form.

### 2.2. Data and Sample Collection

The following data were collected from the participants, including age, height, weight, and BP. Casual spot urine samples were collected from all participants, with disposable urine cups. Then, 2 × 2 mL aliquots of spot urine samples were extracted by trained field investigators. The collection of 24 h urine started immediately after the spot urine collection. In detail, the participants were provided with 6 × 1 L urine containers and instructed to collect all urine voids in these containers during the next 24 h. Participants were requested to return these containers to the village center at around 23.5 h post the starting time and void one last time at the village center. Urine from all containers was thoroughly mixed in a big barrel, and then 2 × 2 mL aliquots of 24 h urine samples were extracted. The start time, finish time, the total urine volume, and the self-reported 24 h missed urine volume were recorded.

For quality control purposes, an additional aliquot (hereafter termed as blind samples) of both spot urine and 24 h urine was collected from 45 randomly selected participants, using the same method. For these additional samples, only the study team was aware of the participants’ identifiers, but the staff at the central lab was not aware of the participants’ identifiers. All urine samples were kept at −20 °C and analyzed in the central laboratory of the Beijing Civil Aviation General Hospital within 7 days.

### 2.3. Laboratory Testing

Urinary sodium (Na) and potassium (K) in the urine were detected by the ion-selective electrode method, and urinary creatinine (Cr) was assayed by the sarcosine oxidase method. These assays were performed on the HITACHI 7600 automated biochemistry analyzer(Hitachi Co., Tokyo, Japan). The standard solution for machine validation was tested for every 100 urine samples. Quality-control measures of the test included calculating the covariance of the standard solution centration and checking the difference between the concentration of the normal urine samples and the blind urine samples. For this analysis, the following urine samples were excluded: (1) the self-reported 24 h missed urine volume was over 20% of the total volume; (2) the total 24 h urine volume was less than 500 mL; (3) the 24 h urinary creatinine excretion (24UCrV) was < 4 mmol or >25 mmol in women or <6 mmol or >30 mmol in men [27]; (4) the 24 h urine volume was more than 3 standard deviations of the population mean [28]; (5) the spot urinary creatinine concentration (spot _Cr_) was more than 3 standard deviations of the population mean.

### 2.4. Sample Size Calculation

According to previous studies [28,29], the correlation coefficients between estimated and measured 24UNaV ranged from 0.30 to 0.70 [28], and the correlation coefficients between estimated and measured 24UKV ranged from 0.19 to 0.71 [29]. For a conservative assumption of 0.32 for the correlation coefficient between the estimated value and the measured value in the new formula, with a significance level of 0.05, a test efficiency of 0.90, and an exclusion rate of 10%, a sample size of 435 pairs of spot urine samples and 24 h urine samples were needed for developing the new formula. This sample size was calculated using PASS software version 15.

### 2.5. Formula Establishment and Validation

A sex-specific equation for 24UNaV and 24UKV was established using the training data by stepwise multivariate linear regression. The equation for estimating 24UNaV included age, body mass index (BMI), spot urinary sodium concentration (spot _Na_), and spot urinary sodium-to-spot creatinine ratio (spot _Na_/spot _Cr)_. The equation for estimating 24UKV included age, BMI, spot urinary potassium concentration (spot _K_), and the natural log(ln)-transformed Ln [spot urinary potassium-to-spot creatinine ratio (spot _K_/spot _Cr_)] and Ln (spot _Na_/spot _Cr_).

There were, in total, 970 pairs of spot urine and 24 h urine samples from Shanxi province. Based on previous studies [17,30], the ratio between the sample size for training the new formula and the sample for testing the new formula was defined as 7:3. Therefore, we randomly split the 970 pairs of spot urine and 24 h urine samples into 70% (689 pairs) for training and 30% (281 pairs) for validation.

### 2.6. Statistical Analysis

The formulas for estimating 24UNaV and 24UKV were estimated from the 281 pairs of testing data. We calculated the means and standard deviations of the measured 24UNaV and 24UKV and the estimated 24UNaV and 24UKV using published formulas and the newly developed formula. The measured and estimated 24UNaV and 24UKV were compared using the paired samples T-test. Meanwhile, the mean bias, the Pearson correlation coefficient (r), and the intraclass correlation coefficient (ICC) of the 24UNaV and 24UKV, between measured and estimated values, were calculated. Using the “2-way mixed single measures test (absolute agreement)”, the ICC was determined with SPSS version 21.0. ICC values between 0.50 and 0.75 indicate moderate reliability, values between 0.75 and 0.90 indicate good reliability, and values greater than 0.90 indicate excellent reliability [31]. Differences between the new formula and the published formulas of the Pearson correlation coefficient were tested with Cocor version 1.10 [32]. Bland–Altman plots were used to further express the agreement between the measured and estimated values from different formulas. At the individual level, the P30 (the proportion of estimates within 30% difference from the measured sodium excretion) [33], the relative differences [(estimated-measured) × 100%/measured] and the absolute differences (estimated-measured) were calculated [34,35].

Continuous variables were expressed as mean ±standard deviations, while categorical variables were described as proportions (%). All statistical analyses were performed using SPSS (version 21.0, SPSS & IBM,Inc, Chicago, IL, USA.), MedCalc(version 17.6, Ostend, Belgium), Prism(version 9.4, GraphPad Software, LLC, San Diego, CA, USA),Cocor (version1.10, Australia), and PASS (version 15, NCSS Statistical software, LLC, USA) software. The level of significance for statistical tests was set at *p* ≤ 0.05.

## 3. Results

### 3.1. Participant Enrollment and Basic Clinical Characteristics

The flowchart of participant enrollment is shown in Figure 1. A total of 1287 participants were initially included. After excluding 317 individuals, data from 970 (75.4%) stroke patients were included in this study. Data from 689 participants were included in the training cohort, and data from 281 participants were included in the testing cohort. The demographic data and clinical characteristics of the study population are shown in Table 1.

### 3.2. Establishment of the New Formula for Estimating 24UNaV and 24UKV

One multivariate linear regression model was developed for estimating 24UNaV and 24UKV for males and females, respectively (Table 2). For both males and females, the correlation coefficients for 24UNaV were positive for spot _Na_, spot _Na_/spot _Cr_, and BMI, and negative for age, while the correlation coefficients for 24UKV were positive for spot _K_, Ln (spot _K_/spot _Cr_), Ln (spot Na/spot Cr), and BMI, and negative for age. The correlation coefficient of 24UNaV was 0.42 for males and 0.33 for females. The correlation coefficient of 24UKV was 0.71 for both males and females.

### 3.3. Validity Comparison of the New Formula and other Published Formulas at Population Level

The published formulas for estimating 24UNaV and 24UKV from spot urine samples are shown in Table 3. Table 4 compares the means and standard deviations, the mean bias, the correlation coefficient, the ICC, and the P30 between the measured and estimated values of 24UNaV and 24UKV.

The mean biases between the measured and estimated values using the new formula were 5.17 mmol/day (95% CI: −1.93, 12.27 mmol/day) for 24UNaV and 0.85 mmol/day (95% CI: −0.80, 2.51 mmol/day) for 24UKV. The values estimated by the new formula showed no significant difference compared to the measured values (*p* = 0.153 for 24UNaV and *p* = 0.312 for 24UKV) (Table 4). The values estimated by the other three published formulas showed significant differences compared to the measured values (all *p* < 0.05) (Table 4).

The correlation coefficients of the measured values and estimated values using different formulas were between 0.35 and 0.45 for 24UnaV, and between 0.62 and 0.71 for 24UKV (all *p*  <  0.05) (Table 4). For all different formulas, the ICC was under 0.35 for estimating 24UNaV and under 0.69 for estimating 24UKV. However, the P30 suggested that the accuracy of the new method was low (59.43% for 24UNaV and 70.11% for 24UKV), but higher than other published formulas (49.47%, 50.53%, and 56.58% by the Kawasaki, INTERSALT, and Tanaka formulas, respectively, for 24UNaV; 38.79% and 59.79% by the Kawasaki and Tanaka formulas, respectively, for 24UKV) (Table 4).

Bland–Altman plots illustrating the mean biases of the four formulas are shown in Figure 2 and Figure 3. The new formula performed relatively accurately among the four formulas. The plots show underestimation at higher excretion levels and overestimation at lower levels.

### 3.4. Comparison of the New Formula and other Published Formulas at the Individual Level

For the estimation of 24UNaV, the new formula, Kawasaki formula, INTERSALT formula, and Tanaka formula had 24.2%, 16.1%, 13.9%, and 18.6%, respectively, estimated values that were within ±10% relative difference from the measured values (Appendix A), and had 23.5%, 14.9%, 19.2%, and 19.2%, respectively, estimated values that were within ±17.1 mmol/day absolute difference from the measured values (Appendix A).

For the estimation of 24UKV, the new formula, Kawasaki formula, and Tanaka formula had 25.6%, 13.2%, and 21.4%, respectively (Appendix A), estimated values that were within ±10% relative difference from the measured values, and had 70.1%, 37.4%, and 62.2%, respectively, estimated values that were within ±12.8 mmol/day absolute difference from the measured values (Appendix A).

## 4. Discussion

The risk of stroke is associated with high sodium intake and low potassium intake [4,5,36]. The accurate detection of sodium and potassium intake is very important for evaluating the risk of stroke [37,38]. However, few special formulas can evaluate their intake in stroke patients, given the inadequate performance of the existing formulas. Herein, our study established a new formula to evaluate 24UNaV and 24UKV for stroke patients, using spot urine samples. Meanwhile, we validated the efficiency of the new formula, and compared it with other published formulas in stroke patients.

The new formula was established by stepwise multivariate linear regression for men and women separately due to the fact that different urinary sodium and potassium excretions have a strong association with biological sex [7,39,40,41]. BMI and age are significantly associated with Na and K excretion in both sexes [39]. Meanwhile, compared with Na and K alone, Na/Cr and K/Cr are independent of urine volume, which may degrade if specimens are not well-protected in a temperature-controlled environment [42]. Therefore, our formula for 24UNaV included the variables of age, BMI, spot _Na_, and spot _Na_/spot _Cr_, and, for 24UKV, consisted of the variables of age, BMI, spot _K_, Ln (spot _k_/spot _Cr_), and Ln (spot _Na_/spot _Cr_). A higher BMI indicates higher food consumption, and higher food consumption reflects higher sodium and potassium intake [39,43]. In our study, the result of regression analyses similarly showed that the correlation coefficients were positive between BMI and 24UNaV (24UKV) in both males and females. We also found that age was negatively associated with 24UNaV and 24UKV in both males and females. This may be because younger individuals may prefer processed or restaurant-prepared foods [39] and may have higher food consumption, which may result in higher salt intake. In our study, there was a different salt intake between the salt substitute group and the regular salt group, but the mean difference in 24 h urinary sodium excretion between the two groups was 12.30 mmol (equivalent to 0.72 g regular salt). The minor difference in salt intake can be ignored for the formula calculation.

The comparison between the three previously published equations and the new formula for estimating 24UNaV and 24UKV in stroke patients showed that these published formulas might be inferior to the new formula. The Tanaka formula was the closest to the actual excretion, and the Kawasaki method exhibited the largest bias. This may be explained by the following two reasons. The mean differences between the estimated values and the measured values differed among studies and populations [44]. The INTERSALT formula was based on the data of 52 centers located in 32 countries from all over the world, with higher BMIs, younger ages, and lower intakes of salt [20], while the Kawasaki method and the Tanaka formula were developed based on data from the Japanese population, with lower BMIs and higher intakes of salt. However, our data was based on Chinese stroke patients, which might explain the lower error in our validation results of 24UNaV and 24UKV. Methodologically, the INTERSALT formula is a linear estimating equation, which may overestimate the low salt zone because there is an intercept [45]. Meanwhile, the Kawasaki formula and the Tanaka formula can estimate 24UNaV and 24UKV based on a similar hypothesis that there is a strong correlation between the measured and estimated values of 24UCrV, and that 24UNaV or 24UKV is directly proportional to the value of Na/Cr or K/Cr in a spot urine sample multiplied by weight, height, and age [14]. As an intermediate step based on the estimation of 24UCrV, the calculation adds imprecision, and the removal of one step should help reduce statistical errors in the estimation procedure [20].

There are two reasons for the different performances of the Tanaka formula and Kawasaki method, although they were based on a similar hypothesis. In our study, the correlation coefficient between the estimated and measured 24UCrV was 0.65 for the Kawasaki formula (*p* < 0.05) and 0.54 (*p* < 0.05) for the Tanaka formula (Appendix A). However, the Kawasaki formula was inferior to Tanaka formula. Firstly, the 24UCrV is significantly lower in women than in men [46]. Compared with the Tanaka formula, the Kawasaki formula estimated 24UCrV for men and women separately, which might be the primary result of the good performance of the estimation for 24UCrV by the Kawasaki formula. However, different from the constant nature within a small range, and lacking a diurnal rhythm like that of urinary creatinine excretion [22,47], urinary sodium and potassium excretions have substantial day-to-day variation [7,15]. Secondly, the difference in the collection time of urine may affect the performance of estimation. In the Kawasaki formula, the second morning voiding urine specimen is used, while in our study, spot urine was randomly collected in the daytime.

The strength of this study was the population-based sample with a reasonably large sample size and high participation rate. In addition, in our study, casual spot urinary samples were collected, which can be easily collected without causing any discomfort and can be easily stored. However, there were several limitations in this study. First, similar to the INTERSALT formula, the new formula was a linear estimating equation, which may overestimate the low salt zone. Second, single 24 h urine samples were collected, and the daily sodium and potassium intake of intra- and inter-individual variations many affect the accuracy of estimated sodium and potassium excretions.

## 5. Conclusions

This study developed a new formula for stroke patients to evaluate 24UNaV and 24UKV. Although the agreement between the reference values and the estimated values from the new method varies with the variation of the measured values of 24UNaV, the new formula was still a meaningful exploration to estimate 24UNaV and 24UKV in stroke patients, which may contribute to the risk management of stroke.

## Figures and Tables

**Figure 1 nutrients-14-04105-f001:**
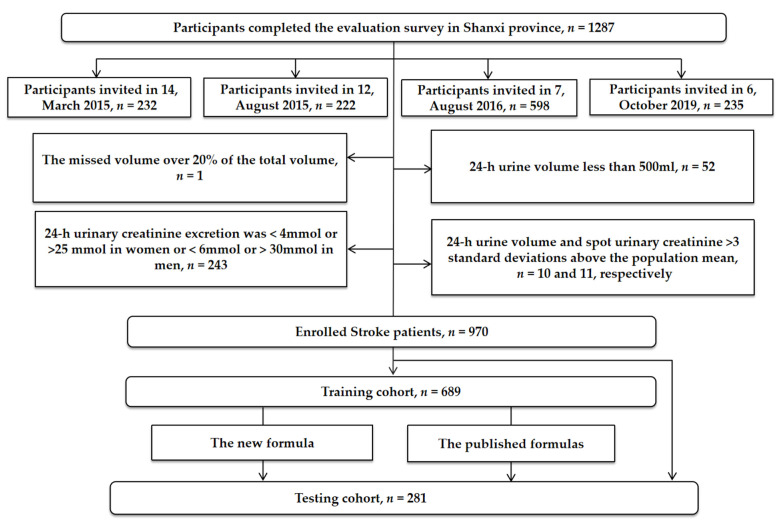
Flowchart for participant enrollment.

**Figure 2 nutrients-14-04105-f002:**
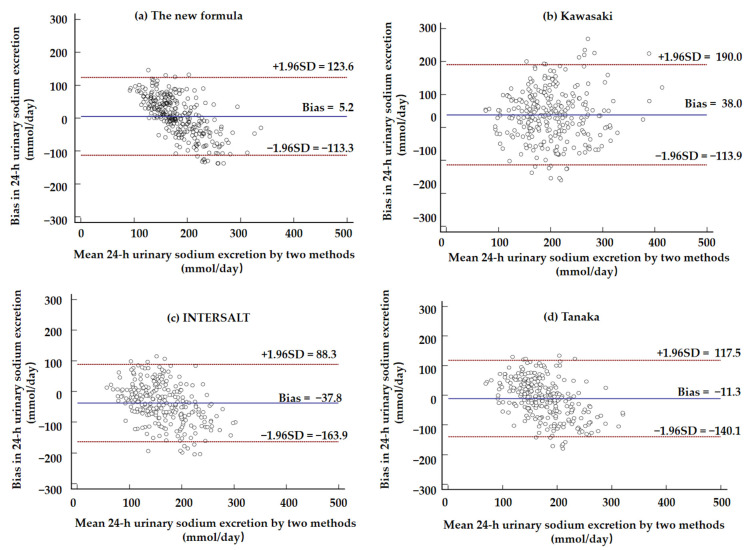
Bland–Altman plots showing the difference in 24 h urinary sodium excretion (24UNaV) between measured and estimated values from the new formula (a), Kawasaki formula (b), INTERSALT formula (c), and Tanaka formula (**d**). The difference was calculated as the estimated values minus the measured values. The middle line is the mean bias, while the upper and lower dashed lines are the upper and lower limits, respectively.

**Figure 3 nutrients-14-04105-f003:**
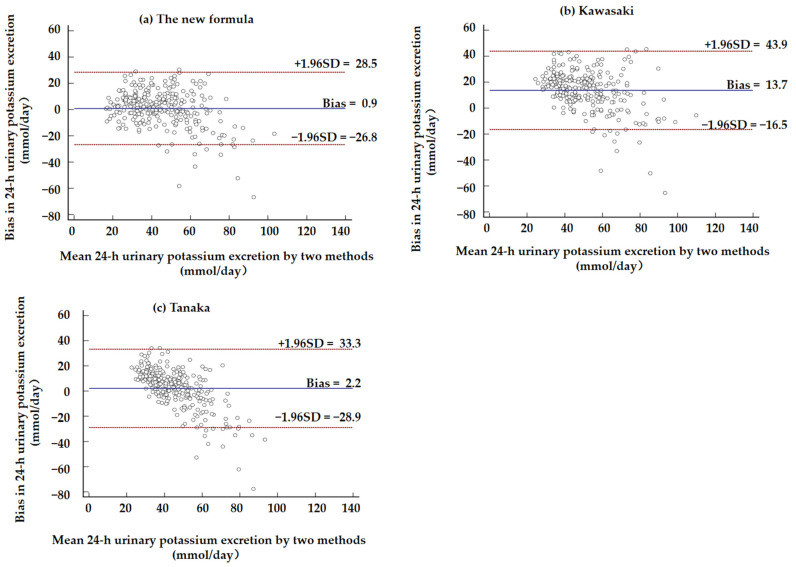
Bland–Altman plots showing the difference in 24 h urinary potassium excretion (24UKV) between measured and estimated values from the new formula (**a**), Kawasaki formula (**b**), and Tanaka formula (**c**). The difference was calculated as the estimated values minus the measured values. The middle line is the mean bias, while the upper and lower dashed lines are the upper and lower limits, respectively.

**Table 1 nutrients-14-04105-t001:** Demographic data and clinical characteristics of the study population.

Characteristics	Mean ± SD, *n*(%)			
	Entire Cohort (*n* = 970)	Training Cohort (*n* = 689)	Testing Cohort (*n* = 281)	*p*-Value
Sex, male	531 (54.74)	389 (56.46)	142 (50.53)	0.093
Age (years)	64.53 ± 7.87	64.43 ± 7.92	64.76 ± 7.75	0.557
BMI (kg/m^2^)	24.85 ± 3.44	24.73 ± 3.39	25.14 ± 3.53	0.086
Systolic BP (mmHg)	146.19 ± 21.32	146.48 ± 21.67	145.48 ± 20.48	0.507
Diastolic BP (mmHg)	85.70 ± 12.90	86.03 ± 13.40	84.87 ± 11.56	0.175
**Spot urinary analysis**				
Potassium (mmol/L)	55.79 ± 36.09	56.90 ± 37.58	53.06 ± 32.05	0.108
Sodium (mmol/L)	131.85 ± 62.88	132.53 ± 63.82	130.16 ± 60.58	0.594
Creatinine (mmol/L)	8.93 ± 5.60	9.05 ± 5.78	8.65 ± 5.15	0.315
**24 h urinary analysis**				
Total urine volume (L)	1.60 ± 6.00	1.61 ± 0.62	1.58 ± 0.56	0.445
Potassium (mmol/day)	45.95 ± 21.02	46.59 ± 21.42	44.38 ± 19.93	0.138
Sodium (mmol/day)	184.94 ± 72.80	186.69 ± 74.79	180.66 ± 67.61	0.242
Creatinine (mmol/day)	8.03 ± 2.56	8.11 ± 2.62	7.85 ± 2.41	0.154

Note: BP, blood pressure; SD, standard deviation; BMI, body mass index.

**Table 2 nutrients-14-04105-t002:** Establishment of the equation for estimating 24UNaV and 24UKV from spot urine by multiple linear regression.

Variables	Male				Variables	Female			
	β	Standard Error	Student’s *t* Test	*p*-Value		β	Standard Error	Student’s *t* Test	*p*-Value
Sodium									
Constant	41.492	44.655	0.929	0.353	Constant	147.159	45.611	3.226	0.001
Age (years)	−0.191	0.450	−0.424	0.672	Age (years)	−1.030	0.512	−2.012	0.045
BMI (kg/m^2^)	4.349	1.120	3.885	0.000	BMI (kg/m^2^)	2.011	1.065	1.887	0.060
Spot _Na_ (mmol/L)	0.229	0.058	3.943	0.000	Spot _Na_ (mmol/L)	0.143	0.066	2.168	0.031
Spot _Na_/spot _Cr_	1.744	0.343	5.085	0.000	Spot _Na_/spot _Cr_	1.035	0.253	4.095	0.000
Potassium									
Constant	−1.035	10.898	−0.095	0.924	Constant	4.318	11.222	0.385	0.701
Age (years)	−0.052	0.100	−0.515	0.607	Age (years)	−0.235	0.114	−2.072	0.039
BMI (kg/m^2^)	0.410	0.249	1.647	0.100	BMI (kg/m^2^)	0.530	0.237	2.239	0.026
Spot _K_ (mmol/L)	0.031	0.032	0.968	0.334	Spot _K_ (mmol/L)	0.040	0.031	1.281	0.201
Ln (spot _K_/spot _Cr_)	33.280	2.330	14.281	0.000	Ln (spot _K_/spot _Cr_)	30.990	2.509	12.352	0.000
Ln (spot Na/spot _Cr_)	−5.789	1.826	−3.170	0.002	Ln (spot _Na_/spot _Cr_)	−7.837	1.762	−4.449	0.000

Note: BP, blood pressure; BMI, body mass index; SD, standard deviation; 24UNaV, 24 h urinary sodium excretion; 24UKV, 24 h urinary potassium excretion; spot _Na_, spot urinary sodium concentration; spot _K_, spot urinary potassium concentration; spot _Cr_, spot urinary creatinine concentration.

**Table 3 nutrients-14-04105-t003:** The published formulas for estimating 24UNaV and 24UKV from spot urine samples.

Method	Urine Sample	Formula for Estimating 24 h Urinary Excretion (mmol/day)
Sodium		
Kawasaki [13]	Second morning urine	16.30 × (spot _Na_ (mmol/L)/spot _Cr_ (mg/L) × PrUCr_24h_ (mg/day)) ^0.5^ PrUCr_24h_(mg/day) = 15.12 × Weight (kg) + 7.39 × Height (cm) − 12.63 × Age (years) − 79.90 (male) PrUCr_24h_(mg/day) = 8.58 × Weight (kg) + 5.09 × Height (cm) − 4.72 × Age (years) − 74.50 (Female)
INTERSALT [20]	Casual spot urine	(25.46 + 0.46 × spot _Na_ (mmol/L)) − 2.75 × spot _Cr_ (mmol/L) − 0.13 × spot _K_ (mmol/L) + 4.10 × BMI (kg/m^2^) + 0.26 × Age (years) (Male) (5.07 + 0.34 × spot _Na_ (mmol/L)) − 2.16 × spot _Cr_ (mmol/L) − 0.09 × spot _K_ (mmol/L) + 2.39 × BMI (kg/m^2^) + 2.35 × Age (years) − 0.03 × Age^2^ (years) (Female)
Tanaka [14]	Casual spot urine	21.98 × (spot _Na_ (mmol/L)/spot _Cr_ (mg/L) × PrUCr_24h_ (mg/day)) ^0.392^ PrUCr_24h_(mg/day) = 14.89 × Weight (kg) + 16.14 × Height (cm) − 2.04 × Age (years) − 2244.45
**Potassium**		
Kawasaki [13]	Second morning urine	7.20 × (spot _K_ (mmol/L)/spot _Cr_(mg/L) × PrUCr_24h_ (mg/day)) ^0.5^ PrUCr_24h_(mg/day) = 15.12 × Weight (kg) + 7.39 × Height (cm) − 12.63 × Age (years) − 79.90 (male) PrUCr_24h_(mg/day) = 8.58 × Weight (kg) + 5.09 × Height (cm) − 4.72 × Age (years) − 74.50 (Female)
Tanaka [14]	Casual spot urine	7.59 × (spot _K_ (mmol/L)/spot _Cr_ (mg/L) × PrUCr_24h_ (mg/day))^0.431^ PrUCr_24h_ (mg/day) = 14.89 × Weight (kg) + 16.14 × Height (cm) − 2.04 × Age (years) − 2244.45

Note: PrUCr_24h_, predictive 24 h urinary creatinine excretion; spot _Na_, spot urinary sodium concentration; spot _K_, spot urinary potassium concentration; spot _Cr_, spot urinary creatinine concentration.

**Table 4 nutrients-14-04105-t004:** Comparison between measured and estimated values of 24UNaV and 24UKV.

Formula	Mean ± SD (mmol/day)	Mean Bias (mmol/day, 95%CI)	Pearson’s r	ICC (95% CI)	P30 (%)
**24UNaV (mmol/day)**					
Measured value	180.66 ± 67.61	Reference	Reference	Reference	Reference
New formula	185.83 ± 32.40	5.17 (−1.93, 12.27)	0.45 a	0.35 (0.24, 0.45) ^a^	59.43
Kawasaki	218.68 ± 70.48	38.02 (28.91, 47.12) ^a,b^	0.37 ^a,b^	0.32 (0.16, 0.46) ^a^	49.47
INTERSALT	142.86 ± 44.64	−37.80 (−45.35, 30.25) ^a,b^	0.40 ^a,b^	0.30 (0.11, 0.46) ^a^	50.53
Tanaka	169.33 ± 41.62	−11.3 (−19.05, −3.61) ^a,b^	0.35 ^a,b^	0.31 (0.20, 0.41) ^a^	56.58
**24UKV (mmol/day)**					
Measured value	44.38 ± 19.94	Reference	Reference	Reference	Reference
New formula	44.89 ± 14.84	0.85 (−0.80, 2.51)	0.71 ^a^	0.69 (0.62, 0.75) ^a^	70.11
Kawasaki	58.04 ± 14.10	13.66 (11.85, 15.47) ^a,b^	0.64 ^a,b^	0.46 (0.05, 0.68) ^a,b^	38.79 ^b^
Tanaka	46.59 ± 9.80	2.21 (0.35, 4.08) ^a,b^	0.62 ^a,b^	0.49 (0.39, 0.57) ^a,b^	59.79

Note: SD, standard deviation; 24UNaV, 24 h urinary sodium excretion; 24UKV, 24 h urinary potassium excretion; Pearson’s r, Pearson correlation coefficient; ICC, intraclass correlation coefficient; 95% Cl, 95% confidence interval. Reference refers to measured value. ^a^ Significant difference/correlation were detected by comparing to the measured value. ^b^ Significant difference/correlation were detected by comparing to the new formula.

## Data Availability

The data presented in this study are available upon request from the corresponding author.

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
