# Peer review of "A Method for Estimating 24 h Urinary Sodium and Potassium Excretion by Spot Urine Specimen in Stroke Patients"

_nutrients, 2022, doi:10.3390/nu14194105_

Round 1

Reviewer 1 Report

The aim of the paper by Beike Wu et al. is to develop and validate a new formula for 24UNaV and 24UKV by spot urine specimen in stroke patients. And They concluded that

 the new formula provides a reasonable method to estimate 27 24UNaV and 24UKV in stroke patients by using spot urine specimen.

My personal feeling is that 24-h urine collection is  the golden standard to evaluate Na and K intake. To be precise, several collections are required to identify the individual urinary Na and K excretion; the single collection, on the other hand, is able to identify the average collection of a population sample  and therefore is a useful tool in epidemiological studies However the paper is interesting and well conducted.

In my opinion there are only minor points that could be raised:

-       The paper requires an English revision

-       Figures 4 and 5 are complicated and add no more informations

-       Intervention trials must be discussed in the paper

-       In the references he authors forget much of the European research  which has played a fundamental role in this area of research:

Introduction :

High Na and low K risk factors for stroke: doi: 10.1016/j.jacc.2010.09.070;

doi: 10.1136/bmj.b4567; doi: 10.1007/s13668-021-00383-z; doi: 10.1016/j.numecd.2018.11.010.

24-h urine collection: doi: 10.1016/j.numecd.2013.06.011.

Discussion:

Association of Na intake with BMI and age: doi:10.1097/HJH.0000000000000010; doi: 10.1038/ejcn.2010.22; doi: 10.1016/j.numecd.2020.10.017

Author Response

Response to Reviewer 1 Comments

Q1:The paper requires an English revision

Response 1: Thank you for your advice. We have polished the language throughout the manuscript.

Q2:Figures 4 and 5 are complicated and add no more information

Response2: Thank you for your comment. We have deleted Figures 4 and 5 from the main part of the manuscript and included them as supplementary material.

Q3: Intervention trials must be discussed in the paper

Response3: Thank you for your comment. We had included the intervention trials in the part of “Study participants ” We have add as following:“The villages were randomly assigned in a 1:1 ratio to the intervention group, in which the participants used a salt substitute (75% sodium chloride and 25% potassium chloride by mass), or to the control group, in which the participants continued to use regular salt (100% sodium chloride).”(Page 2 ,Line 77-80)

Q4:In the references the authors forget much of the European research which has played a fundamental role in this area of research

Response4: Thank you very much for your recommended references, which are valuable and helpful for us to complete and improve our paper. We have added these references as appropriate throughout our revised manuscript.

doi: 10.1016/j.jacc.2010.09.070: 9. D'Elia L, Barba G, Cappuccio FP, Strazzullo P: Potassium intake, stroke, and cardiovascular disease a meta-analysis of prospective studies. J Am Coll Cardiol 2011, 57:1210-1219.

doi: 10.1136/bmj.b4567: 10. Strazzullo P, D'Elia L, Kandala NB, Cappuccio FP: Salt intake, stroke, and cardiovascular disease: meta-analysis of prospective studies. Bmj 2009, 339:b4567.

doi: 10.1007/s13668-021-00383-z: 11. Cappuccio FP, Campbell NRC, He FJ, Jacobson MF, MacGregor GA, Antman E, Appel LJ, Arcand J, Blanco-Metzler A, Cook NR, et al: Sodium and Health: Old Myths and a Controversy Based on Denial. Curr Nutr Rep 2022, 11:172-184.

doi: 10.1016/j.numecd.2018.11.010: 12. Cappuccio FP, Beer M, Strazzullo P: Population dietary salt reduction and the risk of cardiovascular disease. A scientific statement from the European Salt Action Network. Nutr Metab Cardiovasc Dis 2018, 29:107-114.

doi: 10.1016/j.numecd.2013.06.011:15.Ji C, Miller MA, Venezia A, Strazzullo P, Cappuccio FP: Comparisons of spot vs 24-h urine samples for estimating population salt intake: validation study in two independent samples of adults in Britain and Italy. Nutr Metab Cardiovasc Dis 2014, 24:140-147.

doi: 10.1016/j.numecd.2020.10.017:36.Donfrancesco C, Lo Noce C, Russo O, Minutoli D, Di Lonardo A, Profumo E, Buttari B, Iacone R, Vespasiano F, Vannucchi S, et al: Trend of salt intake measured by 24-h urine collection in the Italian adult population between the 2008 and 2018 CUORE project surveys. Nutr Metab Cardiovasc Dis 2021, 31:802-813.

doi:10.1097/HJH.0000000000000010:40. Galletti F, Agabiti-Rosei E, Bernini G, Boero R, Desideri G, Fallo F, Mallamaci F, Morganti A, Castellano M, Nazzaro P, et al: Excess dietary sodium and inadequate potassium intake by hypertensive patients in Italy: results of the MINISAL-SIIA study program. J Hypertens 2014, 32:48-56.

doi: 10.1038/ejcn.2010.22:41 Venezia A, Barba G, Russo O, Capasso C, De Luca V, Farinaro E, Cappuccio FP, Galletti F, Rossi G, Strazzullo P: Dietary sodium intake in a sample of adult male population in southern Italy: results of the Olivetti Heart Study. Eur J Clin Nutr 2010, 64:518-524.

Reviewer 2 Report

The manuscript by Wu et al. presents a new formula that will estimate 24-hr urinary excretion of sodium and potassium in stroke patients using a spot urine specimen. Although an important topic that can throw light on the sodium and potassium status in stroke, I have several concerns that need to be addressed for it to be fit for publication.

Major:

·         Why do we need a separate formula for stroke patients? Why cant Kawasaki or INTERSALT formulae be used? Why are they inadequate for stroke patients?  They have been previously used for hypertensive individuals. In a study by Qian et al (2021), comparison of five formulae showed the INTERSALT method to exhibit goo performance in hypertensive population in China. Need proper justification for the need of the new formula?

·         The sample size calculation showed that 435 cases is a good sample size. What is meant by ‘cases’. My understanding was that all participants of the study had either stroke or hypertension.

·         How were the numbers for ‘testing’ and ‘training’ cohort decided? What was the rationale for deciding 70% for training and 30% for testing.

·         INTERSALT includes more than just North American and European population. It has participants from 52 centers located in 32 countries from all over the world.

Minor:

·         Abstract – the sentence ‘However, the validity …. Using the existing formula” should be rewritten as “However, the validity is limited for 24UNaV and unknown for 24UKV in stroke patients, using the existing formulae”.

·         Materials and Methods: 2.1 Study participants- Why is this study approved by the Institutional Review Board at the University of Sydney, Australia. I don’t understand the relevance here?

·         Table 4: Foot note should indicate what you mean by ‘Reference’.

Author Response

Response to Reviewer 2 Comments

Major:

Q1: Why do we need a separate formula for stroke patients? Why cant Kawasaki or INTERSALT formulae be used? Why are they inadequate for stroke patients?  They have been previously used for hypertensive individuals. In a study by Qian et al (2021), comparison of five formulae showed the INTERSALT method to exhibit good performance in hypertensive population in China. Need proper justification for the need of the new formula?

Response 1: Thank you for your comment. We have updated the introduction of the paper. The main reason for needing a new formula is that previously published estimating equations showed different performances among different populations and may not be adequete for estimating the sodium and potassium intake among patients of stroke.  We have add as follows:“Various prior validation studies indicated that the underlying study population where the validation was conducted is an important factor that affects the performance of these equations [1, 2] For instance, in Asian populations where saltintake is veryhigh, the Kawasaki equation may perform better than the other equations [2, 4]and in Western populations the INTERSALT equations may outperform the others ADDIN EN.CITE [5,6]. Even within the INTERSALT study, the intercepts and coefficient parameters of different population characteristics included in the equations varied for different regions [7]. Therefore these previously published equations may not be adequate to estimate the 24UNaVamong patients of stroke. In addition, there were much fewer studies for estimating 24UKV compared to estimating 24UNaV. ” (Page 2,Line 47-56)

Q2:The sample size calculation showed that 435 cases is a good sample size. What is meant by ‘cases’. My understanding was that all participants of the study had either stroke or hypertension.

Response 2: Thank you for your detailed reading. We agree that cases here maybe confusing. The participants included in this study are at high risk of stroke and yes the majority of them had a history of prior stroke. The “cases” refered to the number of pairs of spot urine and 24-h urine samples required. Accordingly, we have updated the wording to “pairs of spot urine and 24-h urine samples”.(Page 4 ,Line154,157)

Q3:How were the numbers for ‘testing’ and ‘training’ cohort decided? What was the rationale for deciding 70% for training and 30% for testing.

Response 3: Our apologies for not making this clear. We have added as follows:“Based on previous researches such as Peng[4],Sun[8] ,the ratio between the sample size for training the new formula to the sample for testing the new formula was defined as 7:3. Therefore, we randomly split the 970 pairs of spot urine and 24-h urine samples into 70% (689 pairs) for training and 30% (281 pairs) for validation, and the sample size was enough”.(Page 4 ,Line 157-160)

Q4:INTERSALT includes more than just North American and European population. It has participants from 52 centers located in 32 countries from all over the world.

Response 4: Thank you very much for your careful reading and precious suggestion. According your comments,we have corrected it in discuss section as follows:“INTERSALT of 52 centers located in 32 countries from all over the world with higher BMI, younger age and lower intake of salt.”(Page 11 ,Line 311-313)

Minor:

Q5:Abstract – the sentence ‘However, the validity …. Using the existing formula” should be rewritten as “However, the validity is limited for 24UNaV and unknown for 24UKV in stroke patients, using the existing formulae”.

Response 5: Thanks for your valuable suggestion. I have accepted and have added it as follows:“the validity is limited for 24UNaV and unknown for 24UKV in stroke patients, using the existing formulae. ”(Page 1 ,Line 15)

Q6:Materials and Methods: 2.1 Study participants- Why is this study approved by the Institutional Review Board at the University of Sydney, Australia. I don’t understand the relevance here?

Response6: This study uses data from the Salt Substitute and Stroke Study which was sponsored by the George Institute for Global Health Australia. Therefore, ethics approvals were required both from the University of Sydney and Peking University. I have list it in funding section,and add as follows:”SSaSS was sponsored by the George Institute for Global Health Australia”(Page 2 ,Line 70-71)

Q7: Table 4: Foot note should indicate what you mean by ‘Reference’.

Response 7: Thanks for your comments,we have added it in “Table 4”as follows: “Reference refers to measured value”(Page 7,Line 226)

References:

1.Polonia J, Lobo MF, Martins L, Pinto F, Nazare J: Estimation of populational 24-h urinary sodium and potassium excretion from spot urine samples: evaluation of four formulas in a large national representative population. J Hypertens 2017, 35:477-486.

2.Qian N, Jiang Y, Wang Y, Yan P, Yao F, Sun M, Liu X, Zhang Y, Cheng Y, Lu Y, Song W: Validity of five formulas in estimating 24-h urinary sodium via spot urine sampling in hypertensive patients living in Northeast China. J Hypertens 2021, 39:1326-1332.

3.Xu J, Du X, Bai Y, Fang L, Liu M, Ji N, Zhong J, Yu M, Wu J: Assessment and validation of spot urine in estimating the 24-h urinary sodium, potassium, and sodium/potassium ratio in Chinese adults. J Hum Hypertens 2020, 34:184-192.

4.Peng Y, Li W, Wang Y, Chen H, Bo J, Wang X, Liu L: Validation and Assessment of Three Methods to Estimate 24-h Urinary Sodium Excretion from Spot Urine Samples in Chinese Adults. PLoS One 2016, 11:e0149655.

5.Cogswell ME, Wang CY, Chen TC, Pfeiffer CM, Elliott P, Gillespie CD, Carriquiry AL, Sempos CT, Liu K, Perrine CG, et al: Validity of predictive equations for 24-h urinary sodium excretion in adults aged 18-39 y. Am J Clin Nutr 2013, 98:1502-1513.

6.Meyer HE, Johansson L, Eggen AE, Johansen H, Holvik K: Sodium and Potassium Intake Assessed by Spot and 24-h Urine in the Population-Based Tromsø Study 2015-2016. Nutrients 2019, 11.

7.Brown IJ, Dyer AR, Chan Q, Cogswell ME, Ueshima H, Stamler J, Elliott P: Estimating 24-hour urinary sodium excretion from casual urinary sodium concentrations in Western populations: the INTERSALT study. Am J Epidemiol 2013, 177:1180-1192.

8.Sun Y, Wang H, Liang H, Yuan Y, Shu C, Zhang Y, Zhu Y, Yu M, Hu S, Sun N: A Method for Estimating 24-Hour Urinary Sodium Excretion by Casual Urine Specimen in Chinese Hypertensive Patients. Am J Hypertens 2021, 34:718-728.